# Molecular Bases of Drug Resistance in Hepatocellular Carcinoma

**DOI:** 10.3390/cancers12061663

**Published:** 2020-06-23

**Authors:** Jose J.G. Marin, Rocio I.R. Macias, Maria J. Monte, Marta R. Romero, Maitane Asensio, Anabel Sanchez-Martin, Candela Cives-Losada, Alvaro G. Temprano, Ricardo Espinosa-Escudero, Maria Reviejo, Laura H. Bohorquez, Oscar Briz

**Affiliations:** 1Experimental Hepatology and Drug Targeting (HEVEFARM) Group, University of Salamanca, IBSAL, 37007 Salamanca, Spain; rociorm@usal.es (R.I.R.M.); mjmonte@usal.es (M.J.M.); marta.rodriguez@usal.es (M.R.R.); masensio002@usal.es (M.A.); anabelsanchez@usal.es (A.S.-M.); candelacives@usal.es (C.C.-L.); alvarogacho@usal.es (A.G.T.); raespinosa@usal.es (R.E.-E.); maredi@usal.es (M.R.); laurahernandezb@usal.es (L.H.B.); 2Center for the Study of Liver and Gastrointestinal Diseases (CIBERehd), Carlos III National Institute of Health, 28029 Madrid, Spain

**Keywords:** apoptosis, cancer stem cell, DNA repair, epithelial-mesenchymal transition, liver cancer, metabolism, multidrug resistance, refractoriness, transport, tumor environment

## Abstract

The poor outcome of patients with non-surgically removable advanced hepatocellular carcinoma (HCC), the most frequent type of primary liver cancer, is mainly due to the high refractoriness of this aggressive tumor to classical chemotherapy. Novel pharmacological approaches based on the use of inhibitors of tyrosine kinases (TKIs), mainly sorafenib and regorafenib, have provided only a modest prolongation of the overall survival in these HCC patients. The present review is an update of the available information regarding our understanding of the molecular bases of mechanisms of chemoresistance (MOC) with a significant impact on the response of HCC to existing pharmacological tools, which include classical chemotherapeutic agents, TKIs and novel immune-sensitizing strategies. Many of the more than one hundred genes involved in seven MOC have been identified as potential biomarkers to predict the failure of treatment, as well as druggable targets to develop novel strategies aimed at increasing the sensitivity of HCC to pharmacological treatments.

## 1. Introduction

The most frequent (≈90%) primary liver cancer is hepatocellular carcinoma (HCC) which originates from hepatocytes under carcinogenic conditions such as liver cirrhosis. According to the GLOBOCAN database, HCC is the sixth most common and the fourth most deadly cancer, accounting for 840,000 new cases and 780,000 deaths per year worldwide [1].

In spite of emerging immunotherapy being seen as a great hope for the treatment of advanced HCC, at present the outcome of patients that cannot be treated with curative methods (i.e., percutaneous ablation, surgical removal or liver transplantation) still depends on their response to the scarcely efficient available pharmacological armamentarium.

The reason for the high refractoriness of HCC, not only to classical chemotherapy but also to targeted therapy based on the use of inhibitors of tyrosine kinases (TKIs), is still poorly understood. More than one hundred genes have been identified to play a role in the interconnected and sometimes synergistic mechanisms of chemoresistance (MOC) that have been classified into seven groups (from MOC-1 to MOC-7) for a better analysis [2]. This scheme has been used here to review the present knowledge of the clinical impact of MOC on the response to pharmacological strategies currently used to treat HCC patients (Figure 1).

Between 2007 and 2016, sorafenib was the only systemic drug approved for advanced HCC in patients with preserved liver function [3]. Sorafenib is a multi-TKI whose mechanism of action involves the reduction of tumor cell proliferation by blocking RAF serine/threonine kinases in the RAS/RAF/MEK/ERK pathway and angiogenesis through targeting mainly c-KIT, FLT3, VEGFRs, or PDGFR-β pathways [4]. Recently, lenvatinib, another multi-TKI with activity against VEGFR1-3, FGFR1-4, PDGF, RET, and KIT, has been approved as a first-line treatment in the management of advanced HCC patients and has demonstrated similar efficacy to sorafenib [5]. Regorafenib has a molecular target spectrum like that of sorafenib and is an alternative used for HCC patients who are resistant to sorafenib, do not tolerate it, or whose tumor progresses during treatment with this drug. Another novel TKI used as a second-line treatment is cabozantinib, which also inhibits several tyrosine kinases, including VEGFR1-3, MET, and AXL [5].

In addition, several monoclonal antibodies with anti-angiogenesis activity or immune checkpoint inhibitors (ICIs) are being added to the list of approved systemic HCC strategies. Among first-line therapies are nivolumab, an antibody against programmed cell death protein-1 (PD-1), and the combination of atezolizumab (a PD-1 inhibitor) and bevacizumab (an angiogenesis inhibitor also called AtezoBev). Ramucirumab, which specifically binds to the VEGFR2 domain, and pembrolizumab (a PD-1 inhibitor) have also been approved as second-line treatments [5].

Regarding classical chemotherapeutic drugs, transarterial chemoembolization (TACE) with doxorubicin and cisplatin, and less frequently also 5-fluorouracil (5-FU), is the treatment of choice for some patients with advanced HCC [6].

## 2. Drug Uptake and Export (MOC-1)

To become effective, most anti-HCC drugs require certain intracellular levels to be reached, so any change resulting in a reduction of intracellular drug accumulation may compromise the treatment success. Impaired expression or function of plasma membrane proteins involved in the transport of these drugs (collectively referred to as the “transportome”) may constitute part of MOC-1, in which lower drug uptake (MOC-1a) and enhanced drug efflux (MOC1-b) can be distinguished [7].

### 2.1. Drug Uptake Carriers (MOC-1a)

Solute carriers (SLC) play an essential role in the uptake of anticancer drugs. For instance, although a controversial study has reported the inability of organic cation transporter-1 (OCT1, *SLC22A1*) to transport sorafenib [8], others using *Xenopus laevis* oocytes and mammalian cells with stable expression of OCT1 have revealed that this transporter is involved in the uptake of sorafenib [9,10], but not regorafenib [10]. Reduced *SLC22A1* expression and impaired OCT1 function, due to loss-of-function mutations and aberrantly spliced variants in HCC, have been reported (Table 1) [9,10]. Furthermore, the loss of OCT1 expression in the plasma membrane of tumor cells has been correlated with reduced overall survival (OS) of HCC patients treated with sorafenib [11]. Moreover, it has been shown that restoring the OCT1 expression results in enhanced sensitivity of HCC cells to sorafenib [10].

Notably, the organic-anion-transporting polypeptide (OATP) family (*SLCO* genes) may be involved in the transport of TKIs used against HCC. OATP1B1 and OATP1B3 can mediate the uptake of cabozantinib [12], while OATP1B1 can also transport regorafenib [13] and probably lenvatinib [14]. However, there is a controversy regarding OATP-mediated sorafenib transport. Although transport assays with stably transfected cells suggest that sorafenib is a substrate for OATP1B1/1B3 [15], experiments with transgenic mouse models expressing human OATP1B1/1B3 showed that these transporters only have a role in the clearance of sorafenib glucuronidated metabolites [16]. OATP1B1/1B3 are down-regulated in HCC [17], suggesting that these changes may contribute to the chemoresistance of HCC.

SLC transporters may also affect the intracellular fate of anticancer drugs. A recent study has suggested that down-regulation of the SLC46A3 carrier localized at the lysosomal membrane, as other proton-coupled transporters, is involved in sorafenib resistance, and hence can be considered a marker of HCC prognosis and a new therapeutic target [18]. Tumors with lower expression of SLC46A3 had more aggressive phenotypes, and patients had a lower OS. Restoration of SLC46A3 expression in HCC cells decreased their mesenchymal and stemness characteristics and increased their sensitivity to sorafenib [18].

### 2.2. Drug Export Pumps (MOC-1b)

ATP-binding cassette (ABC) proteins are crucial players in multidrug resistance (MDR), because they can transport a wide variety of anticancer drugs, such as TKIs (Table 1) [7]. The up-regulation of ABC proteins in HCC is often associated with the activation of survival pathways (MOC-5b) [33] and the acquisition of mesenchymal or stem cell-like phenotypes by tumor cells (MOC-7) [34].

MDR1 (*ABCB1*), also known as P-glycoprotein, has been associated with reduced median survival time (MST) in HCC [20]. Among MDR1 substrates are sorafenib [35] and regorafenib [36]. Moreover, pharmacological inhibition of MDR1 moderately increased the plasma concentration of lenvatinib, suggesting a possible role in the transport of this drug [37].

Some members of the multidrug resistance-associated proteins (MRP, *ABCC* family), such as MRP2 (*ABCC2*) and MRP3 (*ABCC3*), but not MRP1 (*ABCC1*), contribute to the MDR phenotype in HCC [38]. MRP2 is involved in the transport of sorafenib [39], regorafenib [13], cabozantinib [12], and probably lenvatinib [23], whereas MRP3 expression has been associated with the lack of cultured HCC cell sensitivity to sorafenib [22].

The breast cancer resistance protein (BCRP, *ABCG2*) plays a dominant role in sorafenib efflux [40]. The expression of this pump has been found higher in HCC tissue than in adjacent non-tumor tissue [19]. Increased BCRP expression has also been correlated with reduced OS in HCC patients [19]. Therefore, it has been proposed as a predictor of HCC response to sorafenib [41].

Interestingly, several TKIs can inhibit the activity and expression of ABC pumps leading to collateral sensitivity, which opens a new therapeutic option for TKIs as chemosensitizers in HCC [42]. Thus, sorafenib can down-regulate *ABCB1* and *ABCC2* in HCC [43], probably through the inhibition of survival pathways. Moreover, reduced expression of BCRP by treatment of HCC cells with TKIs, such as gefitinib, increases their sensitivity to sorafenib [41]. Similarly, cabozantinib can be considered as a chemosensitizing agent in HCC due to its ability to inhibit MDR1 function [44].

Besides changes in the expression levels, the presence of genetic variants (GV) of these pumps markedly affects the pharmacological response of HCC. Thus, HCC patients who are heterozygous for variants rs1045642 in *ABCB1,* and rs2231137 and rs2231142 in *ABCG2*, have lower sorafenib plasma levels and better clinical evolution [21]. However, the presence of these variants does not affect the pharmacokinetics of lenvatinib [23].

## 3. Drug Metabolism (MOC-2)

Changes in the expression or activity of enzymes involved in drug metabolism may result in a reduced prodrug activation or an increased drug inactivation, both leading to a lower proportion of the active agent in cancer cells (Table 1). This is, for instance, the case of dihydropyrimidine dehydrogenase (DPD) and thymidine phosphorylase. Expression levels of DPD, the main enzyme in 5-FU catabolism, show a high interindividual variation, which influences the toxicity, resistance, and efficacy of 5-FU [45]. Whereas no relationship between treatment outcome and the expression of thymidine phosphorylase was found, *DPD* expression has been correlated with a poorer prognosis, in terms of disease progression rate (DPR) and progression-free survival (PFS) of HCC patients treated with 5-FU-based TACE, suggesting that high DPD expression could be used as a predictive marker of 5-FU ineffectiveness [25].

However, conflicting results have been reported in HCC patients treated with S-1. This is an orally administered preparation that combines tegafur (a prodrug of 5-FU), 5-chloro-2,4-dihydropyridine (a reversible DPD inhibitor), and oteracil potassium (an inhibitor of orotate phosphoribosyltransferase) used to inhibit 5-FU activity within normal gastrointestinal mucosa and hence reduce its toxic side effect on this tissue. Surprisingly, individuals with high expression of DPD showed improved OS after S-1-based chemotherapy [26]. DPD has been proposed as a therapeutic target for interferon-α (IFN-α) treatment, which is effective in some HCC patients in reducing metastasis by inhibition of epithelial-mesenchymal transition (EMT) through DPD down-regulation. Individuals with higher DPD expression have shown worse response to IFN-α therapy, indicating that DPD might be both a drug target and a prognostic biomarker [27].

Although several studies have addressed the impact of sorafenib metabolism on the prognosis of HCC patients, no clear link between its metabolism and the response to this drug has been established. Sorafenib is metabolized by cytochrome P450 (CYP), more precisely through CYP3A4/5-mediated oxidation and UDP-glucuronosyltransferase 1A9 (UGT1A9)-mediated glucuronidation.

After oral administration, the majority of the drug (≈77%) is recovered in feces (50% as the unchanged compound) due to lack of absorption and biliary excretion, while ≈19% of the administered sorafenib is excreted in the urine, mainly as sorafenib glucuronide. Biliary and renal routes of elimination involve glucuronidation by UGT1A9, whereas other UGTs may account for glucuronidation of oxidized metabolites generated by CYP3A4/5 [46]. Altered pharmacokinetics of its metabolites might affect sorafenib effectiveness and toxicity.

Interindividual variability regarding sorafenib-induced toxicity may be associated with the existence of GVs affecting CYPs and UGTs. In Chinese patients with HBV/HCV-associated HCC, individuals with the *CYP3A5*3* (rs776746) variant displayed minimal sorafenib metabolism together with severe liver and kidney damage [24]. Polymorphisms in both *UGT1A9* (rs3832043) and the *NR1I2* gene encoding the nuclear receptor PXR (rs3814055, rs2472677, and rs10934498), possibly affecting the hepatic expression of UGT and CYP enzymes, have also been linked to high and persistent sorafenib plasma levels and severe toxicity [30]. Another polymorphism of *UGT1A9* (rs17868320) has been associated with diarrhea and early severe toxicity in patients receiving sorafenib [31]. In microsomes obtained from HCC, *CYP3A4* and *UGT1A9* down-regulation has been associated with decreased sorafenib metabolism [47]. Besides, low *UGT1A9* expression has been related to a worse prognosis of HCC patients treated with adjuvant sorafenib. Moreover, in HCC specimens, a significant inverse relationship between miRNA-200a/-183 and *UGT1A9* mRNA levels was observed. Therefore *UGT1A9*, under epigenetic regulation of miRNA-200a/-183, could identify patients who might benefit from adjuvant sorafenib treatment [29].

In addition to being a UGT1A9 substrate, sorafenib is also a potent inhibitor of UGT1A1, the enzyme responsible for bilirubin glucuronidation and biliary detoxification. Some *UGT1A1* GVs have been associated with sorafenib-induced toxicity. For instance, patients carrying *UGT1A1*28* (rs8175347) treated with sorafenib undergo higher hepatic exposure to this drug and hence an increased risk of acute hyperbilirubinemia, which often leads to treatment interruption [28]. Consistently, sorafenib can cause hyperbilirubinemia in patients with Gilbert’s syndrome [48].

Regorafenib and sorafenib share many pharmacokinetic and pharmacodynamic properties. Regorafenib metabolism is comparable with that of sorafenib, as it occurs through oxidative biotransformation, predominantly by CYP3A4 (23% of the administered dose), and glucuronidation by UGT1A9 (18%). Most orally-administered regorafenib is excreted via the biliary/fecal route (mainly remaining as unmodified regorafenib), while a minor amount is excreted into urine [49,50]. In patients with colorectal cancer, the polymorphism *UGT1A9*22* (rs3832043) in the gene promoter is related to the appearance of severe toxic hepatitis after regorafenib treatment, whereas none of the affected patients had *CYP3A4* mutations [32]. Regorafenib is also a potent UGTs inhibitor, which is consistent with the hypothesis that inhibition of UGT1A1 is involved in the hyperbilirubinemia observed in patients treated with this drug [51].

Cabozantinib and lenvatinib share metabolic routes, undergoing hepatic biotransformation by CYP enzymes (mainly, but not only, CYP3A4) and conjugation with glucuronic acid (via UGT1A9) and/or sulfate [52]. The impact of *CYP3A4/5* GVs on the pharmacokinetics of lenvatinib has been reported for the first time in thyroid cancer patients [23]. Particularly, the *CYP3A4*1G* (rs2242480) polymorphism was found to influence steady-state plasma lenvatinib concentrations. Further studies are needed to ascertain the influence of biotransformation of these two therapeutic agents on their efficacy and tolerance in HCC patients.

## 4. Changes in Drug Targets (MOC-3)

Resistance to targeted therapies used against HCC is often caused by reactivation of the signaling pathway inhibited by the drug. This occurs by alterations in upstream or downstream regulatory routes or by secondary modifications of the drug target (Table 2).

Studies with naive and sorafenib-resistant HCC cells showed that the dysregulation of EGFR and HER3 pathways reduced the efficacy of sorafenib [54]. Moreover, the transcription factor KLF4 induced the development of sorafenib resistance in HCC cell lines and cooperated with EGFR to form a positive feedback loop to amplify the resistance to this drug [53].

The detection by immunohistochemistry of high levels of VEGFR-2 and phosphorylated ERK (p-ERK) in tumor tissue can be predictive of poor outcome in advanced HCC patients treated with sorafenib [55]. High levels of phosphorylated MET (p-MET) were also associated with enhanced resistance to adjuvant sorafenib treatment in HCC patients [56]. Moreover, an association between GVs in genes of the angiogenic pathway and the outcome of HCC patients treated with sorafenib has been suggested. For instance, polymorphisms of VEGF and its receptors have been associated with the response to sorafenib. More precisely, those patients with TT or TA genotype of rs1870377 or with GG or GA genotype of rs2305948 in *VEGFR2* receiving first-line treatment with sorafenib presented lower response and shorter PFS, and those with CC genotype for rs2071559 or TT or TA genotype of rs1870377 showed reduced OS [61]. The ALICE-1 study described that the CG genotype of rs2010963 (*VEGFA*) and the CC of rs4604006 (*VEGFC*) were also associated with a worse outcome [59]. These findings have been confirmed by the ALICE-2 study and others, suggesting that the analysis of polymorphisms in *VEGFR*, *VEGFA,* and *VEGFC* can be useful to identify HCC patients less likely to benefit from sorafenib [60,67].

In HCC-derived cells, reduced levels of p-MET have been associated with better antitumor activity of cabozantinib [56], whereas *MET* amplification was associated with higher sensitivity to this drug [57].

The Phase III REFLECT study, which compared the efficacy of lenvatinib and sorafenib in unresectable HCC, found that higher serum levels of VEGF, angiopoietin-2 (*ANG2*), and FGF21 were associated with a worse OS in both arms, and increased FGF21 was predictive of a reduced OS in patients treated with sorafenib [58].

Regarding the targets of ICIs, it should be considered that PD-1 is present in CD4^+^ and CD8^+^ T cells and natural killer cells. The association of the antibodies to PD-1 prevents the binding of PD-1 with its ligands, PD ligand 1 (PD-L1), and 2 (PD-L2), present in tumor cells and other hepatic cells and leucocytes, and hence allows a strong T-cell response toward HCC cells [68]. An important proportion of patients do not respond to ICIs, but there is still limited information on the underlying mechanisms accounting for this refractoriness. Low PD-L1 expression has been associated with lower response rates and survival in patients with other types of cancer [69,70]. However, the analysis of PD-L1 expression in CheckMate040 and Keynote224 trials showed no association between immunohistochemical detection in HCC and response to treatment with ICIs [71,72].

## 5. DNA Repairing (MOC-4)

MOC-4 encompasses cellular strategies involving DNA repairing processes (Table 2). At least five major DNA repair pathways that can be involved in HCC chemoresistance have been described: nucleotide excision repair (NER), base excision repair (BER), homologous recombination (HR), non-homologous end joining (NHEJ), and mismatch repair (MMR) [2,73].

Key elements of the NER mechanism, such as the excision repair cross-complementing proteins (ERCC) and the product of the *Xeroderma pigmentosum* (*XP*) group genes, are involved in the repair of DNA adducts caused by alkylating agents such as cisplatin. Immunohistochemical analyses have shown high expression of ERCC1 in ≈50% of HCCs and lower sensitivity to cisplatin in surgically resected HCC tissues with increased expression of ERCC1 [62]. However, these expression data are different from those obtained in other HCC cohorts, whose patients displayed poor ERCC1 expression [74]. Therefore, it is unclear if this protein could play a major role in the lack of response to cisplatin in HCC patients. 

Regarding ERCC5, its up-regulation has been correlated with a worse prognosis after surgery [75], but its relationship with chemoresistance has not been investigated. Among the group of *XP* genes, *XPC* is overexpressed in HCC [76], but its link with the response to the chemotherapy used against HCC has not been studied yet.

RUVBLs are ATPases involved in some chromatin remodeling complexes. Some members of this family are responsible for the detection and repair of DNA damage. RUVBL1 and RUVBL2 (also known as pontin and reptin, respectively) promote cell proliferation in vitro. Both are up-regulated in HCC, which has been associated with poor prognosis [77,78]. Although RUVBL2 depletion in vitro has been shown to increase the sensitivity of HCC cells to genotoxic agents, its relevance in HCC chemoresistance is not known [79]. Regarding RUVBL1, in silico analysis of HCC samples from the public TCGA database reveals that its expression correlates with glucose metabolism and mTOR signaling, which is consistent with a role in hepatocyte proliferation and hepatocarcinogenesis [80]. In healthy cells, these ATPases are located strictly in the nucleus, but in tumor cells, they are also present in the cytoplasm. This dual localization seems to be necessary for tumor cell survival and tumorigenicity [77], which leads to shorter survival of patients with HCC [81].

BER mechanism involves the tightly coordinated function of four kind of enzymes that correct single base modifications that do not distort DNA helix (glycosylase, apurinic/apyrimidinic endonuclease –APE1–, polymerase, and ligase) [73]. The association of the rs25487 polymorphism of X-ray repair cross-complementing protein 1 (XRCC1), one of the main factors of BER mechanism, with the risk of HCC is well known [82]. Moreover, patients with the wild-type genotype (GG) receiving TACE containing a platinum derivative had a worse prognosis than patients carrying AA or GA genotypes [64]. Contradictory results regarding the carcinogenic risk of another *XRCC1* polymorphism, rs1799782, have been reported [83,84]. Nevertheless, this polymorphism has been associated with cisplatin resistance in HCC [65]. In addition, the rs1130409 polymorphism of *APE1*, another enzyme belonging to the BER family, has been clearly associated with the development of resistance to cisplatin [65]. Furthermore, in vitro and in vivo studies revealed that APE1 is related to resistance to radiotherapy [85]. The rs1052133 variant of 8-oxoguanine DNA glycosylase 1 (*OGG1*), another BER gene, is associated with an increased risk of HCC [86]. Besides, OGG1 overexpression in mitochondria of HCC cells enhances the sensitivity to cisplatin, because an imbalance in the activity of elements involved in BER results in the generation of more DNA damage and increased cell death [87].

XRCC4 and XLF (XRCC4-like factor) facilitate the joining of double-chain ends as part of the specific mechanism of NHEJ [88]. Data from TCGA and AMC databases, as well as from the analysis of samples from patients and cell lines, reveal a dramatic XRCC4 up-regulation after treatment with TACE containing platinum drugs, which suggests that this chemotherapy can induce its overexpression. Not surprisingly, patients with higher XRCC4 expression have a worse outcome [63]. Consistently, HCC patients treated with TACE (doxorubicin plus cisplatin) with low expression of XRCC4 had better prognosis [66].

The expression of ataxia telangiectasia mutant (ATM) protein kinase, which participates in the mechanisms of DNA repair through the HR pathway of double strand break repair, is also altered in HCC. Blocking ATM activity in HCC cell lines results in an enhanced antitumor effect of sorafenib, through the inhibition of the AKT pathway (MOC-5) [89].

The MMR system is the mechanism involved in the correction of mismatched nucleotides. The loss of some of its key elements, such as human mutL homolog-1 (hMLH1) and human mutS homolog-2 (hMSH2), is responsible for the so-called “microsatellite instability” [90]. The role of defective DNA MMR in HCC is controversial. Some authors have reported that defective MMR does not contribute significantly to hepatocellular carcinogenesis [91], whereas others have shown that low expression of hMLH1 and hMSH2 is related to tumor progression, enhanced chemoresistance and hence poor prognosis [92].

## 6. Balance between Pro-Survival and Pro-Apoptotic Factors (MOC-5)

### 6.1. Pro-Apoptotic Factors (MOC-5a)

The avoidance of apoptosis activation due to the impaired function of pro-apoptotic proteins is one of the critical mechanisms accounting for the inadequate pharmacological response of HCC (Table 3). Inactivating and gain-of-function (GOF) mutations in essential genes that occur in ≈30% of HCCs are considered to be drivers of tumor progression [93], higher tumor recurrence rate [94], and poor prognosis [95].

The well-known protein p53 is the main tumor suppressor responding to stress signals by transcriptional regulation of genes involved in cell cycle arrest, DNA repair, apoptosis, and senescence. In HCC, its function is often compromised, which is mainly due to the presence of mutations that affect the DNA-binding domain of the protein [96]. Inactivation of p53 in HCC can lead to chemoresistance by suppressing the apoptotic pathways and acquiring a stem-cell-like phenotype [95]. Impaired expression of p53-interacting proteins such as SIRT1 and nucleostemin also contributes to sorafenib resistance [97,98]. Since the most common mutated forms of p53 are considered undruggable [93], several strategies have been proposed to restore wild-type p53 activity and to sensitize HCC to antitumor drugs [99].

The *CDKN2B* gene encodes p15^INK4b^, a cyclin-dependent kinase inhibitor that regulates cell growth by preventing the activation of CDK4/6 kinases. Down-regulation of *CDKN2B* reduces sorafenib-induced apoptosis in HCC cells [100]. Furthermore, this gene is clinically relevant since decreased *CDKN2B* expression is associated with a poor prognosis of HCC patients treated with sorafenib [100].

The p53 up-regulated modulator of apoptosis (PUMA) is a pro-apoptotic member of the BCL-2 protein family involved in the mitochondrial apoptosis pathway activated by sorafenib [122] and cabozantinib [123]. Targeted knock-down of PUMA using specific siRNAs inhibited sorafenib-induced apoptotic features in HCC cells [103]. PUMA-mediated apoptosis induced by sorafenib requires signaling by the NF-κB pathway and the intervention of GSK3β, which are also often dysregulated in HCC [124].

The expression of BMF, another member of the BCL-2 family involved in the activation of the apoptosis downstream effector caspase-3, is inhibited by miR-221 [101]. In addition, *CASP3*, which encodes caspase-3, is also a direct target of this miRNA [102]. The enhanced miR-221 expression is associated with a more aggressive HCC phenotype leading to lower OS and time to progression (TTP) [101]. Thus, circulating serum levels of miR-221 may predict the response to sorafenib in patients with HCC [102].

### 6.2. Survival Pathways (MOC-5b)

Constitutive activation of survival pathways leads tumor cells to prevent apoptosis and therefore plays an essential role in the lack of response to antitumor drugs (Table 3). This is, for instance, the case of the Wnt/β-catenin pathway, which is often aberrantly activated in HCC [125]. Inflammatory signals from the tumor microenvironment, such as transforming growth factor-beta (TGF-β), promote the loss of E-cadherin, which normally keeps β-catenin retained at the plasma membrane, resulting in the accumulation of β-catenin in the cytoplasm and nucleus and allowing oncogenic transcription [126]. In addition, 40–60% of HCCs have mutations in the *CTNNB1* gene, some of which produce β-catenin resistant to degradation in the proteasome, or in *AXIN1* or *AXIN2* genes, destabilizing the β-catenin degradation complex [105,127]. Increased levels of β-catenin in the cytoplasm and nucleus have been found in more than half of HCCs, which have been associated with shorter PFS [128]. β-catenin activates transcription of genes that increase resistance to sorafenib-mediated apoptosis, such as *MYC* [107], *JNK* [112], *BCL2*, and *MCL1* [106].

Increased activity of the PI3K/AKT/mTOR signaling pathway, which is a common event in HCC affecting ≈50% of tumors, has been correlated with poor prognosis, early recurrence, and reduced OS [129]. Overactivated PI3K/AKT/mTOR pathway prevents tumor cells from entering sorafenib-induced apoptosis, which results in resistance to this drug [117]. Inactivating mutations of *PTEN*, an important negative regulator of this pathway, induce hyperactivation of the signaling cascade [130]. Moreover, increased phosphorylation of AKT [131] and RPS6 [118] are common in HCC. It has also been proposed that sorafenib may activate the PI3K/AKT pathway through the induction of SNAI1, resulting in the acquisition of secondary resistance to this TKI by HCC cells [119]. Due to the importance of the PI3K/AKT pathway in the development of resistance to sorafenib, numerous strategies targeting this pathway have been proposed to sensitize HCC cells to this drug [129].

The GOF mutations in *JAK1* that occur in HCC cause the constitutive activation of the JAK/STAT3 pathway, resulting in faster disease progression [111]. Increased STAT3 activity has also been observed in sorafenib-resistant HCC cells [120].

The dysregulation of the Hippo pathway in HCC plays an important role in the development of resistance to sorafenib [121]. The main effector of this pathway is YAP, which shows aberrant overexpression and nuclear localization in ≈50% of HCC [132]. This justifies the increasing interest in finding inhibitors of this pathway to sensitize HCC to sorafenib [129].

The MAPK/ERK pathway has been found overactivated in HCC cell lines resistant to TKIs due to the presence of genomic alterations in receptors, kinases (NRAS), and RAS inhibitors (NF1 and RSK2) [57]. An increased amount of p-ERK has been proposed as a prognostic marker of the response to sorafenib in HCC [116].

Notch and Hedgehog signaling pathways play an essential role in stem cell self-renewal and cell fate determination and they have been proposed as targets to overcome sorafenib resistance in HCC too. A study demonstrated that blocking Notch3 signaling sensitizes HCC cells to sorafenib by down-regulation of p21 and up-regulation of GSK3β [115]. Moreover, in HCC patient-derived organoids enriched in CD44^+^ cells and showing overactivation of Hedgehog signaling, the effect of sorafenib increased when they were incubated with inhibitors of this pathway [110].

ARID1A is a component of the chromatin-remodeling complex that plays a dual role in HCC, as an early-stage oncogene and as a suppressor of advanced-stage tumors [133]. The presence of driver mutations in *ARID1A* is a significant limitation for TKI-based therapy [104]. However, ARID1A deficiency in advanced HCC activates angiopoietin-2-dependent angiogenesis and promotes tumor progression but confers greater sensitivity to sorafenib [134].

Overexpression of FGF19 is common in HCC due to the amplification of the *FGF19* locus at 11q13 [57]. Aberrant signaling through FGF19 and its receptor FGFR4 has been associated with resistance to sorafenib-mediated apoptosis due to inhibition of drug-induced reactive oxygen species production and promotion of EMT through down-regulation of E-cadherin [108].

## 7. Adaptation to Tumor Microenvironment (MOC-6)

The stromal tissue surrounding tumor cells constitutes a barrier that makes it difficult for drugs to reach them. Furthermore, hypoxia, acidification of the extracellular medium, production of inflammatory signals, or reactive oxygen species contribute to changes in the tumor microenvironment that can reduce the effectiveness of chemotherapy [2].

Owing to its accelerated growth, HCC requires a high rate of oxygen supply. Thus, although HCC is highly vascularized, this is one of the most hypoxic tumors [135]. Hypoxia has been associated with increased invasiveness and poorer prognosis of HCC due to increased activity of hypoxia-induced transcription factors (HIFs) (Table 4) [136]. In HCC, a relationship between hypoxic microenvironment and sorafenib resistance has been described [137]. This drug decreases intratumor microvessel density leading to enhanced hypoxia, which subsequently triggers adaptive HIF-mediated responses [137]. Accordingly, HIFs are potential targets for HCC therapy and have been proposed as predictors of response. Thus, the ALICE-2 study describes the rs12434438 variant (GG genotype) of *HIF1A* as associated with a worse response and shorter PFS in HCC patients receiving sorafenib [60]. In vitro studies have shown that HIF-1 activation causes overexpression of ABC proteins, such as MDR1 [138], protects against drug-induced apoptosis through *MCL1* and *BIRC3* up-regulation [139], and contributes to survival by inducing autophagy [140].

High expression levels in HCC of the angiogenic mediator annexin A3 (*ANXA3*) correlated with the activation of autophagy and predicted a worse response to sorafenib [141]. Moreover, the combination of an anti-ANXA3 antibody with sorafenib or regorafenib slowed tumor growth and increased survival in vivo [141]. 

Fibrosis is a hallmark in HCC since most of these tumors develop in cirrhotic liver tissue. Cancer-associated fibroblasts (CAFs) and activated hepatic stellate cells (HSCs), which are part of the tissue surrounding the tumor, produce cytokines and growth and angiogenic factors that affect tumor progression and can also modify tumor response to therapy by activating survival pathways [153]. Lysophosphatidic acid (LPA), a pleiotropic growth-factor-like lysophospholipid produced by autotaxin, seems to play also a role in liver fibrosis and HCC [154]. In a multicellular tumor spheroid model, increased expression of collagen 1A1 (*COL1A1*) in HSC was associated with more compact spheroids and higher resistance to sorafenib [142]. Moreover, HSCs secrete laminin-332, an extracellular matrix protein that decreases sorafenib-induced apoptosis by binding to tumor cells via α3β1 integrin and preventing focal adhesion kinase (FAK) ubiquitination, as demonstrated in Hep3B HCC-derived cells [143].

HCC is also characterized by chronic inflammation. Inflammatory cells that are part of the tumor microenvironment contribute to tumor progression, suppression of adaptive immunity and, by releasing different types of signals, can modify drug response [155].

Tumor-associated macrophages (TAMs) constitute a significant component of leukocyte infiltrate. Using subcutaneous and orthotopic mouse models of HCC, it has been demonstrated that a hypoxic environment increased the expression in TAMs of triggering receptor expressed on myeloid cells 1 (TREM-1) and the recruitment of Treg lymphocytes, resulting in immunosuppression and resistance to anti-PD-L1 therapy [148]. TAMs play a role in anti-PD-L1 resistance by associating with an increased PD-L1 expression and TAM infiltration in tissues from HCC patients with high expression of osteopontin [145]. Moreover, using mice with chemically induced liver tumors, it has been shown that the combination of anti-PD-L1 and an inhibitor of the osteopontin-dependent pathway in TAMs prolonged mice survival [145].

Tumor-associated neutrophils (TANs) are also important components of HCC stroma. In vivo experiments revealed enhanced TANs infiltration in response to sorafenib treatment, which induced the intratumor infiltration of macrophages and Treg cells by secreting cytokines CCL2 and CCL17 [144]. The depletion of TANs enhanced the response to sorafenib, suggesting that TANs may promote sorafenib resistance. Besides, HCC samples from patients treated with sorafenib before surgery contained more TANs than those obtained from patients without pharmacological treatment [144].

Some cytokines may affect therapy effectiveness. TNF-α is an important inflammatory cytokine mainly produced by macrophages. High expression of TNF-α was associated with a weaker response in HCC patients receiving adjuvant sorafenib therapy [147]. TGF-β contributes to sorafenib resistance by promoting the acquisition of mesenchymal and stemness phenotypes by tumor cells [156] and up-regulating several tyrosine kinase receptors [146].

The secretion of extracellular vesicles, another critical element related to the HCC microenvironment, increases under conditions of stress, such as hypoxia, nutrient deficiency, or exposure to cytotoxic agents [157]. Exosomes derived from HCC cells can induce sorafenib resistance in vitro by activating HGF/MET/AKT signaling pathway and inhibiting in vitro and in vivo sorafenib-induced apoptosis [150]. Moreover, the expression of the long intergenic non-coding RNA ROR (linc-ROR) and its enrichment in exosomes play a functional role in HCC resistance to sorafenib through activation of the TGF-β pathway [149].

Even under normoxia conditions, tumor cells favor glycolysis over mitochondrial function for energy production (Warburg effect). This is accompanied by other metabolic reprogramming in which microenvironmental factors have a substantial impact. NANOG is involved in mitochondrial metabolic reprogramming of cells aiming to supply pro-survival growth signals and adapt to sorafenib-induced both hypoxia and altered glucose metabolism [152]. Gankyrin is a small protein overexpressed in different cancers, including HCC, that drives metabolic reprogramming through *MYC* up-regulation. In patient-derived xenograft tumors with high gankyrin levels, in vitro and in vivo experiments have demonstrated that c-MYC inhibition synergized with sorafenib and regorafenib effects, which suggests that gankyrin can contribute to drug resistance in HCC [151].

## 8. Phenotypic Transition (MOC-7)

EMT causes the loss of differentiation and polarity in epithelial cells, which acquire mesenchymal characteristics, such as increased migratory behavior, invasiveness, metastasis, and resistance to apoptosis activation. Thus, in general, EMT is a process that promotes the progression of HCC to a more malignant phenotype with a worse prognosis [158]. Besides, liver cancer stem cells (LCSCs), which share some phenotypic characteristics with cells that have undergone EMT, may also appear within the tumor [159]. LCSCs can originate from: (i) liver stem/progenitor cells through to the acquisition of oncogenic mutations that neutralize the normal proliferation restrictions present in healthy stem cells; and (ii) mature hepatocytes, whose phenotype is reprogrammed and dedifferentiated in response to the inflammatory microenvironment and the accumulation of mutations during carcinogenesis [160].

Although LCSCs and cells undergoing EMT contribute to the cellular heterogeneity within the tumor, these transformed cells share common genetic signatures, such as high expression of cell adhesion surface glycoproteins (CD44, CD133, CD13, CD24, CD90, EpCAM, and N-cadherin), aldehyde dehydrogenase 1A1 (ALDH1A1), keratin-19 (KRT19), and transcription factors (SNAI1, SLUG, TWIST1, ZEB1, and ZEB2) [161,162,163]. One of the consequences of phenotypic diversity is that each cell subpopulation may have a different degree of sensitivity to TKIs. Some cells, such as side population (SP) cells, are sensitive to sorafenib [112]. This drug can even block the HGF-mediated EMT of HCC cells [164]. However, the stemness and mesenchymal characteristics acquired by HCC cells contribute to the primary resistance to TKIs, and hence antitumor in HCC patients are active mainly on non-stem cancer cells, but often have limited therapeutic effects on LCSCs [165]. High expression of CD133, CD90, CD24, or CD44 is associated with a worse outcome of sorafenib-treated HCC patients (Table 5) [166,167,168]. These membrane glycoproteins, which play a role in cell–cell interactions, are also involved in intracellular signaling networks. The overexpression of these stemness markers causes resistance to sorafenib-induced apoptosis by *BCL2* up-regulation, enhanced kinase activity (AKT, AMPK, and mTOR) [169,170], and increased expression of genes involved in survival pathways (Wnt/β-catenin, Notch, and Hedgehog) [171]. CD44 is also involved in MDM2-mediated p53 inhibition [172]. In addition, CD24 induces sorafenib resistance by increasing autophagy through the AKT/mTOR pathway dysregulation [173]. High expression of CD44 and CD133 in HCC cells also results in the overexpression of ABC transporters [34,174].

EpCAM, associated with the expression of stemness genes, together with alpha-fetoprotein (AFP) expression, have been proposed to define different HCC phenotypic groups [183]. Among them, EpCAM^+^/AFP^+^ (hepatic stem cell-like) and EpCAM^−^/AFP^+^ (hepatocytic progenitor-like) groups have been associated with increased drug resistance due to enhanced cell survival mainly through overactivation of the Wnt/β-catenin pathway [183]. Consistently, the exposure of HCC patient-derived cells to sorafenib resulted in an enrichment in EpCAM^+^ cells, which could contribute to the development of acquired sorafenib resistance [175].

Typical LCSCs and EMT markers have been found in a subset of HCC with poor prognosis. These tumors have a highly invasive, metastatic and sorafenib-resistant phenotype, which is associated with up-regulation of ABC pumps, high expression of KRT19, and overactivation of the TGF-β/SMAD pathway [176,184]. The latter is essential in EMT and LCSCs formation [156]. TGF-β causes nuclear accumulation of β-catenin, which in turn produces a loss of epithelial markers and overexpression of stem markers [126]. The TGF-β pathway increases the expression of tyrosine kinase receptors, such as IGF1R, EGFR, PDGFβR, and FGFR1 in HCC cells, which counterbalances sorafenib’s ability to induce apoptosis [146]. High plasma levels of TGF-β1 in patients with advanced HCC have been associated with a poor response to sorafenib [185] and regorafenib [105]. A recent phase II study has shown that galunisertib, a TGFβR1 inhibitor, administered together with sorafenib, prolonged the OS in HCC patients [186]. Higher levels of SMAD2 and SMAD4, which are signal transducers and transcriptional modulators of the TGF-β pathway, have also been found in patients with recurrent tumors [177].

A key feature of EMT in HCC is the disruption of E-cadherin/β-catenin complexes at cell boundaries, accompanied by nuclear translocation of β-catenin, which ultimately promotes EMT and resistance to sorafenib. TWIST1, a transcription factor encoded by an oncogene that is a target of STAT3, AKT, and Wnt/β-catenin pathways, down-regulates E-cadherin and has been found to be overexpressed in sorafenib-refractory HCC patients [180]. Some long non-coding RNA (lncRNA) such as HOTAIR also promote resistance to sorafenib through down-regulation of E-cadherin [182].

Interestingly, some LCSCs and tumor cells that have acquired a mesenchymal phenotype are not refractory to sorafenib, but after prolonged exposure to the drug, they undergo clonal evolution and become sorafenib-resistant. The drug acts favoring the selection of resistant clones through the up-regulation of *NANOG*, *SOX2*, and *OCT4* [33,152,178]. OCT4 participates in the chemoresistance of HCC by activating the OCT4-TCL1-AKT-ABCG2 axis [33]. OCT4 and SOX2 can reactivate oncofetal proteins such as HLF [178]. The exposure of HCC cells to sorafenib also increases AKT activity through a mechanism involving TSC2 [179]. HLF and TSC2 have been proposed as prognostic markers for the response to sorafenib in HCC [178,179].

Some non-coding RNAs are involved in resistance to TKIs through EMT modulation [187]. Thus, miR-216a and miR-217 can induce EMT by targeting *PTEN* and *SMAD7* and, consequently, activate PI3K/AKT and TGF-β pathways, respectively. The overexpression of these miRNAs has been associated with early tumor recurrence [181]. miR-125b expression has negatively correlated with EMT, stemness, and sorafenib resistance [177]. Plasma levels of miR-125b have also been proposed as predictors of OS in HCC patients treated with regorafenib [105].

Increased expression of MALAT1 has been associated with a worse outcome of HCC patients [113]. This lncRNA mediates resistance to sorafenib through the up-regulation of Aurora-A kinase, which is involved in cell cycle activation.

Autophagy is a complex process that can either increase or decrease the sensitivity of HCC cells to sorafenib, depending on the tumor cell context. Inhibition of autophagy due to activation of the AKT pathway by elevated expression of SNHG1 lncRNA may reduce the sensitivity of HCC cells to sorafenib [188]. In contrast, other lncRNAs overexpressed in HCC such as NEAT1 and HANR may promote resistance to sorafenib through the induction of autophagy [109,114].

## 9. Conclusions and Perspectives

In this review, we have summarized the considerable amount of information generated by the current highly active investigations in the field of liver cancer pharmacology. Of note, more than sixty proteins are involved in the lack of response of HCC to sorafenib. This knowledge is improving our understanding of the complex and dynamic process of drug resistance, which is essential to define new biomarkers with high selectivity and sensitivity that may be useful to predict the failure of the pharmacological treatment before starting drug administration to HCC patients. Moreover, it is equally crucial to identify the weakest points in the HCC front line of defense against the available drugs to move towards the development of novel strategies aimed at sensitizing cancer cells to pharmacological treatment. The most promising lines of action to overcome HCC refractoriness to pharmacological treatment are the combination of (i) improved immunotherapeutic agents aimed at more specifically and effectively sensitizing the immune system against HCC cells without affecting normal hepatocytes, (ii) the synthesis of better TKIs with improved pharmacokinetics and lower systemic toxicity, and (iii) the rational application of sensitizing tools using both pharmacological and gene therapy approaches. The advances in this field are expected to help us to develop novel therapeutic tools that may substantially improve the outcome of patients with advanced HCC during the next decade.

## Figures and Tables

**Figure 1 cancers-12-01663-f001:**
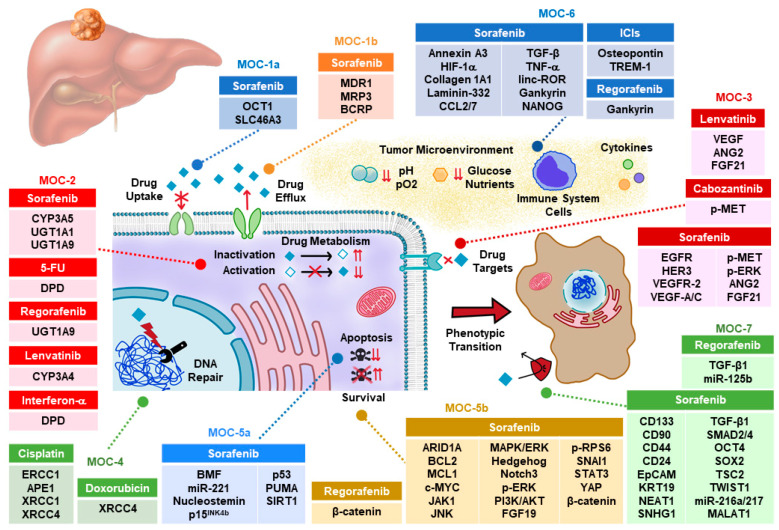
Proteins and non-coding RNAs accounting for chemoresistance in hepatocellular carcinoma. ICI, immune checkpoint inhibitors; MOC, mechanism of chemoresistance.

**Table 1 cancers-12-01663-t001:** Mechanisms of chemoresistance type 1 (MOC-1) and 2 (MOC-2) to drugs clinically used in HCC (hepatocellular carcinoma).

Protein	Change	Drugs Affected	Consequences	Reference
**Uptake Carriers *(MOC-1a)***				
OCT1	Down-regulation	Sorafenib	Reduced OS	[11]
OCT1	Mutations		Decreased function in vitro	[9]
SLC46A3	Down-regulation		Reduced OS	[18]
**Export Pumps *(MOC-1b)***				
BCRP	Up-regulation	Sorafenib	Reduced OS	[19]
MDR1	Up-regulation		Reduced MST	[20]
MDR1	GV: rs1045642		Better clinical evolution	[21]
MRP3	Up-regulation		Decreased cell sensitivity in vitro	[22]
**Drug Metabolism *(MOC-2)***				
CYP3A4	GV: rs2242480	Lenvatinib	Altered plasma levels	[23]
CYP3A5	GV: rs776746	Sorafenib	Hepatic and renal toxicity	[24]
DPD	Up-regulation	5-Fluorouracil	Higher DPR and lower PFS	[25]
DPD	Up-regulation	S-1	Increased OS	[26]
DPD	Up-regulation	Interferon-α	Reduced OS	[27]
UGT1A1	GV: rs8175347	Sorafenib	Hyperbilirubinemia and toxicity	[28]
UGT1A9	Down-regulation		Reduced OS	[29]
UGT1A9	GV: rs3832043		Severe toxicity	[30]
UGT1A9	GV: rs17868320		Severe toxicity	[31]
UGT1A9	GV: rs3832043	Regorafenib	Severe toxicity	[32]

DPR, disease progression rate; GV, genetic variant; MST, median survival time; OS, overall survival; PFS, progression-free survival.

**Table 2 cancers-12-01663-t002:** Mechanisms of chemoresistance type 3 (MOC-3) and 4 (MOC-4) to drugs clinically used in HCC.

Protein	Change	Drugs Affected	Consequences	Reference
**Drug Targets *(MOC-3)***				
EGFR	Positive feedback EGFR-KLF4	Sorafenib	Reduced sensitivity (in vitro)	[53]
EGFR, HER3	Increased activity		Reduced sensitivity (in vitro and in vivo)	[54]
p-ERK, VEGFR-2	Up-regulation		Reduced OS	[55]
p-MET	High levels	Cabozantinib	Increased sensitivity in vitro and in vivo	[56]
p-MET	High levels	Sorafenib	Reduced clinical response	[56]
p-MET	Gene amplification	Cabozantinib	Increased sensitivity in vitro	[57]
VEGF, ANG2, FGF21	High serum levels	Sorafenib, Lenvatinib	Reduced OS	[58]
VEGF-A, VEGF-C	GV: rs2010963, rs4604006	Sorafenib	Reduced OS and PFS	[59,60]
VEGFR-2	GV: rs2071559, rs1870377		Reduced OS, PFS and TTP	[61]
**DNA Repairing *(MOC-4)***				
ERCC1	Up-regulation	Platinum derivatives	Lower sensitivity in surgically resected tissue	[62]
XRCC4	Up-regulation		Reduced OS and PFS	[63]
XRCC1	GV: rs25487		Reduced MST	[64]
XRCC1, APE1	GV: rs1799782, rs1130409		Reduced clinical response	[65]
XRCC4	Down-regulation	Doxorubicin, Cisplatin	Increased OS and PFS	[66]

GV, genetic variant; MST, median survival time; OS, overall survival; p-ERK, phosphorylated ERK; p-MET, phosphorylated MET; PFS, progression-free survival; TTP, time to progression.

**Table 3 cancers-12-01663-t003:** Mechanisms of chemoresistance type 5 (MOC-5) to drugs clinically used in HCC.

Factor	Change	Drugs Affected	Consequences	Reference
**Pro-Apoptotic Factors *(MOC-5a)***				
BMF	Down-regulation	Sorafenib	Reduced OS and TTP	[101]
miR-221	Up-regulation		Reduced OS and TTP	[101]
miR-221	High serum levels		Increased DPR	[102]
Nucleostemin	Up-regulation		Reduced sensitivity (in vitro)	[98]
p15^INK4b^	Down-regulation		Lower survival rate	[100]
p53	Mutations		Reduced OS	[95]
PUMA	Down-regulation		Reduced sensitivity (in vitro)	[103]
SIRT1	Up-regulation		Reduced sensitivity (in vitro)	[97]
**Survival Pathways *(MOC-5b)***				
ARID1A	Mutations	Sorafenib	Reduced OS	[104]
β-catenin	GOF mutations	Regorafenib, Sorafenib	Reduced OS and TTP	[105]
BCL2, MCL1	Up-regulation	Sorafenib	Reduced sensitivity (in vitro)	[106]
c-MYC	Up-regulation		Reduced sensitivity (in vitro)	[107]
FGF19/FGFR4	Increased activity		Reduced sensitivity (in vitro)	[108]
HANR	Up-regulation		Reduced sensitivity (in vitro and in vivo)	[109]
Hedgehog pathway	Increased activity		Reduced sensitivity (in HCC patient-derived organoids)	[110]
JAK1	GOF mutations		Increased DPR	[111]
JNK	Up-regulation		Reduced sensitivity (in vitro)	[112]
MALAT1	Up-regulation		Reduced OS	[113]
MAPK/ERK pathway	Increased activity		Reduced sensitivity (in vitro)	[57]
NEAT1	Up-regulation		Reduced OS	[114]
Notch3	Up-regulation		Reduced sensitivity (in vitro)	[115]
p-ERK	Increased levels		Reduced sensitivity (in vitro and in vivo)	[116]
PI3K/AKT pathway	Increased activity		Reduced sensitivity (in vitro)	[117]
p-RPS6	Increased levels		Increased recurrence rate	[118]
SNAI1	Up-regulation		Reduced sensitivity (in vitro)	[119]
STAT3	Increased activity		Reduced sensitivity (in vitro)	[120]
YAP	Up-regulation		Reduced sensitivity (in vitro)	[121]

DPR, disease progression rate; GOF, gain-of-function; OS, overall survival; p-ERK, phosphorylated ERK; TTP, time to progression.

**Table 4 cancers-12-01663-t004:** Mechanisms of chemoresistance type 6 (MOC-6) to drugs clinically used in HCC.

Factor	Change	Drugs Affected	Consequences	Reference
**Hypoxia**				
Annexin A3	Up-regulation	Sorafenib	Reduced OS	[141]
HIF-1α	Up-regulation		Reduced OS and DFS	[136]
HIF-1α	GV: rs12434438		Reduced OS and TTP	[60]
**Fibrosis**				
Collagen 1A1	Up-regulation	Sorafenib	Reduced sensitivity (in vitro)	[142]
Laminin-332	Up-regulation		Reduced sensitivity (in vitro)	[143]
**Immune System and Inflammation**				
CCL2, CCL17	Up-regulation	Sorafenib	Reduced OS and TTP	[144]
Osteopontin	Up-regulation	ICIs	Reduced sensitivity (in vivo)	[145]
TGF-β	Up-regulation	Sorafenib	Reduced sensitivity (in vitro)	[146]
TNF-α	Up-regulation	Sorafenib	Reduced OS and PFS	[147]
TREM-1	Up-regulation	ICIs	Reduced OS and DFS	[148]
**Extracellular Microvesicles**				
linc-ROR	Up-regulation	Sorafenib	Reduced sensitivity (in vitro)	[149]
HCC-derived Exosomes	High levels		Reduced sensitivity (in vitro and in vivo)	[150]
**Metabolic Reprogramming**				
Gankyrin	Up-regulation	Sorafenib, Regorafenib	Reduced sensitivity (in vitro and in vivo)	[151]
NANOG	Up-regulation	Sorafenib	Reduced sensitivity (in vitro and in vivo)	[152]

DFS, disease-free survival; GV, genetic variant; ICI, immune checkpoint inhibitor; OS, overall survival; PFS, progression-free survival; TTP, time to progression.

**Table 5 cancers-12-01663-t005:** Mechanisms of chemoresistance type 7 (MOC-7) to drugs clinically used in HCC.

Factor	Change	Drugs Affected	Consequences	Ref.
**Cell Adhesion Proteins**				
CD133, CD90	Up-regulation	Sorafenib	Reduced PFS	[166]
CD133, CD44	Up-regulation		Reduced sensitivity (in vitro and in vivo)	[174]
CD44	Up-regulation		Reduced sensitivity (in vitro and in vivo)	[167]
CD24	Up-regulation		Reduced sensitivity (in vitro)	[173]
EpCAM	Up-regulation		Reduced sensitivity (in vivo)	[175]
**Cytokeratins**				
KRT19	Up-regulation	Sorafenib	Reduced sensitivity (in vitro)	[176]
**TGF-β Pathway**				
TGF-β1	Up-regulation	Sorafenib	Reduced sensitivity (in vitro)	[146]
TGF-β1	Up-regulation		Reduced OS and PFS	[105]
SMAD2/4	Up-regulation		Reduced sensitivity (in vitro)	[177]
**Transcription Factors**				
OCT4	Up-regulation	Sorafenib	Reduced sensitivity (in vitro and in vivo)	[33]
OCT4, SOX2	Up-regulation		Reduced sensitivity (in vitro and in vivo)	[178]
TSC2	Increased activity		Reduced sensitivity (in vitro and in vivo)	[179]
TWIST1	Up-regulation		Reduced sensitivity (in vitro and in vivo)	[180]
**Non-Coding RNAs**				
miR-216a/217	Up-regulation	Sorafenib	Reduced DFS	[181]
miR-125b	Down-regulation	Regorafenib	Reduced OS	[177]
HANR	Up-regulation	Sorafenib	Reduced sensitivity (in vitro and in vivo)	[109]
HOTAIR	Up-regulation		Reduced sensitivity (in vitro)	[182]
MALAT1	Up-regulation		Reduced OS	[113]
NEAT	Up-regulation		Reduced OS	[114]

DFS, disease-free survival; OS, overall survival; PFS, progression-free survival.

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
