# Peer review of "Molecular Bases of Drug Resistance in Hepatocellular Carcinoma"

_cancers, 2020, doi:10.3390/cancers12061663_

Round 1

Reviewer 1 Report

The manuscript by Marin et al. is a well-articulated and structured review which offers a clear overview of some of the molecular mechanisms of chemoresistance (MOC) against currently used drugs in hepatocellular carcinoma (HCC) therapy. The manuscript is well-written, clear and easy to read. However, I would recommend some minor revisions before the manuscript can be accepted for publication. The following suggestions should help the authors improve their manuscript.

The authors, after having mentioned drugs used in HCC treatment several times, give a brief description of their mechanisms of action on page 6, lines 186-188, and page 6 line 207. My suggestion is to briefly describe the targets and mechanisms of action for each drug (sorafenib, regorafenib and cabozantinib) in the introduction, where they should also describe what ICIs are and specify which of the many ICIs available today are mentioned in this review.

In addition, the authors should specify how different drugs are currently used in anti-HCC therapy, i.e. whether they are first-line or second-line drugs. It might seem obvious to research workers in the HCC field, but it is not necessarily familiar to others. For an updated review on this topic see the article PMDI 32018226.

The numbering of the references must be corrected. On page 4, line 100, the reference number skips from number 26 to 37, up to 40. The numbering starts again with reference n. 29 (line 120), while reference n. 28 is after n. 29.

Page 4, I suggest a revision of the punctuation from line 122 to 125 to make the sentence clearer.

Page 7, section 5, “DNA repairing (MOC-4)”, should begin with a brief description on the major DNA repair pathways and very briefly describe the major molecules involved in each pathway.

Page 9, in section 6.2, “Survival pathways (MOC-5b)”, the authors should consider and discuss another important pro-survival pathway in HCC, namely the FGF19/FGFR4 signaling pathway, and its role in sorafenib resistance. See the following articles PMDI 32313144 and PMDI 28069043 for example.

I would also recommend considering and discussing the role of long non-coding RNA (lncRNA) in drug resistance in HCC. See these more recent articles as an example: PMID 32462038, PMID 32220970, PMID 32210579, PMID 32103983 and PMID 31549407, but many others could be also mentioned.

The full terms for abbreviations, such as ATM, hMLH1, TREM-1, KRT19, etc., should be given at their first appearance.

Reviewer 2 Report

In this study, Marin et al provide a very comprehensive and state-of-the art review on drug resistance in hepatocellular carcinoma (HCC). Some minor concerns have been raised that probably will improve further the review.            

Minor concerns 

  1. Introduction: I would suggest adding before the first paragraph some general epidemiological issues regarding HCC prevalence and difficulties in surveillance and management because of late diagnosis.

  1. Adaptation to tumor microenvironment (MOC-6): For completeness, I would add in this section a paragraph on autotaxin (ATX). As you know, ATX is a secreted lysophospholipase D that catalyzes the production of lysophosphatidic acid (LPA), a pleiotropic growth-factor-like lysophospholipid. A recent study has published recently on the potential relationship of ATX and liver fibrosis and HCC (Kaffe et al, Hepatology 2017). So, it would be nice to refer and comment on this although there are no clear-cut evidence of a mechanism of drug resistance.

  1. Conclusion and perspectives: I think this section should be lengthened a little giving some more specific directions for the problem.
